# Lead Corrosion and Corrosivity Classification in Archives, Museums, and Churches

**DOI:** 10.3390/ma15020639

**Published:** 2022-01-15

**Authors:** Milan Kouřil, Tereza Boháčková, Kristýna Charlotte Strachotová, Jan Švadlena, Tomáš Prošek, Kateřina Kreislová, Pavlína Fialová

**Affiliations:** 1Department of Metals and Corrosion Engineering, Faculty of Chemical Technology, University of Chemistry and Technology, Technická 5, 166 28 Prague, Czech Republic; tereza.bohackova@vscht.cz (T.B.); strachok@vscht.cz (K.C.S.); 2Department of Metallic Construction Materials, Technopark Kralupy of the University of Chemistry and Technology, Nám. G. Karse 7, 278 01 Prague, Czech Republic; svadlenj@vscht.cz (J.Š.); prosekt@vscht.cz (T.P.); 3SVÚOM s.r.o., U Měšťanského Pivovaru 934/4, 170 00 Prague, Czech Republic; kreislova@svuom.cz (K.K.); fialova@svuom.cz (P.F.)

**Keywords:** indoor corrosivity, volatile organic acids, lead corrosion, preventive conservation

## Abstract

Sixteen localities were involved in a broad study, resulting in the classification of the indoor corrosivity of metals considered in the ISO 11844 standard, especially lead. Recently, lead has been added to the standard as a metal specifically sensitive to volatile organic compounds such as acetic acid. Data on one-year exposure in museum depositories and exhibition spaces, archives, libraries, and churches show that the currently valid lead corrosivity categories are not correctly defined. The obtained data allowed for the proposal of new realistic ranges of indoor corrosivity categories for lead. The exposure program was also used to validate techniques for determining the corrosion degradation of metal coupons. Mass increase and mass loss techniques were supplemented with the galvanostatic reduction technique and the measurement of color changes. The study identified the limitations of the mass gain method. Not only is the galvanostatic reduction technique applicable for silver and copper coupons, but the build-up of reducible lead corrosion products depends on air corrosivity. CIELab color-change measurement has proven to be a simple and easy-to-apply method for monitoring the corrosivity of indoor atmospheres with regard to lead. A more reliable response is provided by the determination of color change after 3 months of exposure rather than after one year.

## 1. Introduction

Museums, libraries, galleries, and other cultural heritage institutions hold many objects of cultural importance in their collections, for which preventive conservation procedures or conservation interventions are needed in order to eliminate degradation processes and ensure their preservation for the future. In general, having milder indoor atmosphere conditions, especially a stable temperature and humidity, and lower concentrations of airborne pollutants, such as dust or nitrogen and sulfur oxides, are helpful in this regard. On the other hand, concentrations of some pollutants, including volatile organic compounds, can increase significantly indoors, in particular volatile organic acids, such as acetic acid and formic acid [1]. They are released into the air from the exhibits themselves (e.g., by paper degradation [2,3]), but also from packaging materials [4], wooden furnishings [5,6,7], and cleaning agents or adhesives [4,7,8,9]. Hardwoods, especially oak, are very dangerous in this respect [8,10,11]. Therefore, the presence of volatile organic acids indoors is often monitored, as evidenced by a number of works dealing with such measurements in real conditions [2,12,13], with the monitoring of emissions of organic substances from different materials and objects depending on the temperature, humidity, and age of the source object [3,11,14,15,16,17,18,19,20].

Volatile organic acids can cause the corrosion of copper, zinc, nickel, iron, and cadmium [7,21,22,23], but among the historical materials represented in collections, lead is the metal most sensitive to the presence of volatile organic compounds in the environment [24]. While it corrodes very slowly under normal atmospheric conditions because of the formation of well-adhering protective films on the surface (most often carbonate-based), the action of volatile organic acids causes the disruption and dissolution of these layers and the subsequent precipitation of incoherent bulky corrosion products based on formate or acetate. These can be transformed by the action of airborne carbon dioxide into more stable carbonate-based corrosion products (cerussite and hydrocerussite), but freed volatile organic acids can enter the corrosion process again. The released acids can concentrate in the surface layer and cause further corrosion, regardless of whether their original source is still present in the environment or not [5,25,26].

The corrosion aggressiveness in the indoor environment can be determined based on ISO 11844, which defines corrosion hazards in low-corrosive environments by measuring environmental parameters [27] or by determining the corrosion rate of selected metals [28]. The standard is also applied in the cultural heritage sector and is used by many museums and other cultural institutions around the world to assess potential corrosion risk and carry out appropriate procedures to protect metal objects. This standard proposes the use of passive samplers, onto which the acids are adsorbed, to measure the concentration of acetic and formic acids in the air by ion chromatography analysis [27]. Commercially available products, such as colorimetric diffusion tubes and chemical impregnated papers, are widely used by cultural institutions. Since pollutant concentrations depend on color changes, the evaluation is very simple and fast. The disadvantages are usually a high detection limit and low accuracy [8]. Due to the former, the samplers require longer exposure, in the order of days to weeks. The result then represents an average over a given period and not an actual value. In the field of cultural heritage care, it is problematic for institutions to carry out more precise and real-time instrumental analyses. This would involve allocating a considerable amount of money to purchase the instrument and train an operator, or to outsource the measurements.

When determining the corrosivity of the indoor atmosphere based on the corrosion of selected metals, metal coupons made of carbon steel, copper, zinc, silver, or lead are exposed to the environment. Lead was introduced to the standard during the last revision in 2020, as it is the material most sensitive to indoor contamination by volatile organic acids. Based on the mass changes (mass increase and mass loss), corrosivity classes IC1–IC5 for each metal were defined, with corresponding corrosion rate intervals [29]. For the adequate application of lead as a specific indicator of contamination with volatile organic acids (VOAs), a large dataset on lead corrosion in environments with different parameters needs to be obtained, and these data need to be linked with data on the corrosion behavior of other standard metals.

From the preventive conservation point of view, it is also desirable to set VOA concentration limits for the storage and display of collection items containing lead. However, a single detection of VOAs may not be representative of exposure conditions. The release rate of VOAs depends on exposure conditions, particularly the temperature and the air exchange rate, and may thus be significantly influenced by seasonal factors, such as central heating and ventilation. The rate of lead corrosion is a direct reflection of environmental corrosivity, as opposed to the instantaneous concentration of VOAs. The advantage of using lead to detect corrosion is that simple assessment techniques, such as mass change or change in appearance, can be used. However, it is necessary to know the relationship between these parameters and the long-term corrosion rate of lead, which includes climatic effects throughout the year.

Lead and other metals used for air corrosivity assessment (silver, copper, and zinc) were exposed for one year at 16 sites in museums, depositories, archives, libraries, and churches. The objective of this study was to compare the available methods for determining lead corrosion rates and the applicability of currently valid indoor atmospheric corrosivity categories for lead. The corrosion rate was assessed by mass gain, mass loss, and electrochemical reduction. The latter technique was used not only for silver and copper, where its use is well established, but also now for lead. A further aim was to evaluate the usefulness of assessing the evolution of lead’s visual appearance using the CIELab color-change measurement as an easy-to-apply method for monitoring the corrosivity of the indoor atmosphere.

## 2. Materials and Methods

### 2.1. Selection of Exposure Locations

The exposure sites were selected to cover a wide range of conditions that are considered with regard to conservation and indoor conditions. Therefore, among them were churches with a completely uncontrolled internal atmosphere interacting with the external atmosphere, rooms with mineralogical collections where increased pollution was expected but there were no fluctuations in temperature and relative humidity, museum depositories and exhibition spaces with controlled atmospheres but the possibility of cryptoclimates being created, and archive or library spaces. Some sites were chosen purely to cover a wide range of conditions, and in some cases the choice was related to the exposure of lead-containing materials, specifically papal bulls, letterpress types, and organ pipes.

University library, room space (location 1): Well-controlled conditions with temperature that does not depend much on sun irradiation through windows because the windows are oriented to the north. Temperature depends more on the action of the central heating system. Relative humidity is not controlled but low relative humidity is expected. Since wooden bookshelves and furniture with large surface area are installed in the room, increased VOA content was expected. The room is attended approximately once a day, and thus the air is exchanged quite frequently;University library, wooden bookcase (location 2): The bookcase is located inside location 1. The same conditions apply, except that the atmosphere inside the bookcase is not exchanged frequently (approximately once a month) and the VOA content might be increased by emission from books;University exhibition hall, room space (location 3): The exhibition contains a collection of minerals, metals, and natural chemical compounds, most of which are enclosed in oak showcases. The air quality is expected to be affected by volatile compounds emitted by wood, glue, paper, and the minerals. Temperature depends on the action of the central heating system in winter. Since the room is located under the roof, temperature rises much higher than 30 °C in summer. Relative humidity is not controlled but low relative humidity is expected. The room is attended approximately once per month;University exhibition hall, wooden showcase (location 4): The showcase is located inside location 3, thus the same temperature conditions apply. There are no minerals in the showcase, but elevated VOA content is expected since the showcase is not ventilated at all;Monastery library, open bookcase (location 5): Partially controlled conditions of large baroque library open to guided tours several times per day. Temperature and relative humidity are not well-controlled. VOA emission from wood and paper is expected. The indoor air might be also contaminated from intensive traffic outdoors;Modern depository of a local museum, room space (location 6): Temperature and relative humidity are controlled. The depository stores archeological objects. No significant VOA emitters are present. Low VOA content is expected;Historical depository of a local museum, room space (location 7): Temperature and relative humidity are controlled. The depository contains wooden furniture and textile and wooden objects. Low VOA content is expected;Historical archive, room space (location 8): Temperature and relative humidity are well-controlled by means of central air conditioning. The air is purified by filters. Stable conditions and low VOA content are expected, although the archive is full of cartons;Historical archive, historical archive box (location 9): Temperature and relative humidity are controlled by means of central air conditioning, but the air enclosed in the box might be affected by VOA emission from the cartons, which contain no deacidification admixtures;Historical archive, modern archive box (location 10): Temperature and relative humidity are well-controlled by means of central air conditioning. The air enclosed in the box is not exchanged, but should not be affected by VOA emission from the carton, since it contains deacidification admixtures and its inner surface is covered with acid-free paper;Technical museum exhibition hall, wooden case (location 11): Temperature and relative humidity are well-controlled by means of central air conditioning. The air is not purified by filters. Inside the case, the air is not being exchanged and its VOA content might be elevated by emission from wood;Technical museum depository, metallic case (location 12): Temperature and relative humidity are controlled by means of central air conditioning. The air is not purified by filters. Inside the metallic case, the air is not being exchanged and its VOA content might be elevated by emission from cellulose acetate film;Technical museum depository, room space (location 13): Temperature and relative humidity are well-controlled by means of central air conditioning. The air is not purified by filters. Low emission of VOAs or other pollutants is expected from technical (mostly electronic) objects stored in the depository;Baroque church, open space (locations 14–16): No control over the indoor conditions is present. Temperature and relative humidity alter with outdoor conditions through broken windows. As a result, the atmosphere is well ventilated and accumulation of VOAs emitted by wooden furniture and organ parts is not expected.

### 2.2. Corrosion Coupons

The choice of materials for corrosivity determination at each site followed the ISO 11844 standard. Silver and copper are widely accepted metals for monitoring corrosion aggressivity, not only by mass changes but also by electrochemical reduction. Recently, lead has been involved in the standard as a metal specifically sensitive to volatile organic compounds like acetic acid. The applicability of using the electrochemical reduction of lead corrosion products formed during exposure under indoor conditions to determine the extent of lead corrosion is not well-established. One of the original metals adopted by ISO 11844 is zinc. Its corrosion rate should also reflect the corrosivity affected by the presence of volatile organic compounds. For comparison, with the sensitivity of lead to such contamination, zinc was included in the exposure.

For the production of silver coupons, silver sheet with 99.99% purity was used (Safina, a.s., Prague, Czech Republic). Copper and zinc sheets with a purity of at least 99.5% were supplied by Ferona, a.s., Prague, Czech Republic. Lead coupons were made from sheet with a purity of 99.97% (Kovohutě Příbram nástupnická, a.s., Příbram, Czech Republic).

Coupon dimensions were set to ensure that their weight met the requirements of the ISO 11844 standard valid prior to the 2020 revision, in view of the determination of mass increase and mass loss using ultra-microbalances with a weighing accuracy of 0.1 µg, thus limiting the sample weight to 2 g. Coupons with a thickness of 0.5 or 1 mm and dimensions of 10 mm × 25 mm or 50 mm were used.

### 2.3. Methods

Indoor corrosivity was assessed using corrosion coupons exposed for one year. The coupons were installed in a vertical position, suspended from a plastic rack when space conditions allowed, or laid horizontally in an appropriate confined space (Figure 1). In both cases, the area of both sides of the coupon was considered in the corrosion rate calculation. The edges and hole sides were neglected. The corrosion coupons were made of metals recommended by ISO 11844 for assessing the corrosivity of indoor atmospheres: silver, copper, zinc, and lead. The corrosion rate was evaluated based on mass increase and mass loss and the amount of corrosion products determined by galvanostatic reduction. Color changes were monitored on the corrosion coupons during exposure. Exposure conditions were monitored using Testo 174 H temperature and relative humidity loggers (Testo SE & Co. KGaA, Lenzkirch, Germany). The content of VOAs, specifically acetic acid, was monitored.

#### 2.3.1. Volatile Organic Acid Content

Passive dosimeter tubes (PDTs) from GASTEC Corp. (model 81D, Ayase, Japan) were used for quick determination of the presence of volatile organic acids. Air diffuses passively through the tube and reacts with the detecting reagent within, giving a color change indicated on the printed scale, providing a multiplication product of VOA content and exposure time. Average VOA content within an exposure period of usually 10 days might be calculated from the reading. The tubes have a relatively low detection limit of 5 ppm·hour.

The long-term average content of VOAs was detected by means of passive samplers produced by the authors. The VOAs were collected for 30 days in triplicate. Exposure of a new set of 3 passive samplers for another 30 days followed. Adsorbed acids were then analyzed by means of ion chromatography. The system is described elsewhere [30].

#### 2.3.2. Determination of Corrosion Rate from Mass Increase

The corrosion rate from the mass increase was determined in accordance with ISO standard 11844-2 [28]. The rate of mass increase *r_mi_*, in milligrams per square meter per year of exposure (mg m^−2^ a^−1^), is calculated as
(1)$$r_{mi}=\frac{m_{ae}-m_{be}}{A\cdot t}$$
where *m_ae_* is the mass of the test specimen in relation to the reference balance standard after exposure (mg), *m_be_* is the mass of the test specimen in relation to the reference balance standard before exposure (mg), *A* is the surface area (m^2^), and *t* is the exposure time (years, with the unit symbol *a*).

Each test specimen was weighed twice (UYA 2.4Y Ultra-Microbalance, Radwag, Radom, Poland) in relation to a reference balance standard of stainless steel with a similar mass to the specimen. The difference between the first mass of the test specimen m_1_ and the reference balance standard *m_r_*_,1_ is calculated as (*m*_1_ − *m_r_*_,1_), and the difference between the second mass pair is calculated in the same way (*m*_2_ − *m_r_*_,2_). The mass of the test specimen is calculated in relation to the reference specimen as the average of the mass differences:(2)$$m=\frac{\left(m_{1}-m_{r,1}\right)+\left(m_{2}+m_{r,2}\right)}{2}$$
where *m* is the mass of the test specimen in relation to the reference balance standard, *m*_1_ is the mass of the test specimen at first weighing, *m*_2_ is the mass of the test specimen at second weighing, *m_r_*_,1_ is the mass of the reference balance standard at first weighing, and *m_r_*_,2_ is the mass of the reference balance standard at second weighing (all in milligrams).

#### 2.3.3. Determination of Corrosion Rate from Mass Loss

ISO 11844-2 recommends removing corrosion products from exposed specimens by a single-step chemical pickling process and eliminating the mass loss of metal due to coupon exposure in the pickling bath from the corrosion rate calculation by determining the mass loss of an unexposed but otherwise identical blank specimen during an identical pickling cycle. The corrosion rate *r_corr_* (mg m^−2^ a^−1^) is then calculated as
(3)$$r_{corr}=\frac{\left(m_{be}-m_{ap}\right)-\left(m_{bp\text{-}blank}-m_{ap\text{-}blank}\right)}{A\cdot t}$$
where *m_be_* is the mass of the specimen in relation to the reference specimen before exposure, *m_ap_* is the mass of the specimen in relation to the reference specimen after pickling, *m_bp_*_-_*_blank_* is the mass of the blank specimen in relation to the reference specimen before pickling, and *m_ap_*_-_*_blank_* is the mass of the blank specimen in relation to the reference specimen after pickling (all in milligrams).

The determination of individual mass values follows the procedure described above and defined by Equation (2).

Samples were pickled according to the procedure recommended in ISO 11844-2 or other validated procedures [31]:Silver: 750 mL hydrochloric acid (HCl, density = 1.18 g mL^−1^) (Penta s.r.o., Prague, Czech Republic) and demineralized water to make up 1000 mL.Copper: 10% H_2_SO_4_ (Lach-Ner, s.r.o., Neratovice, Czech Republic).Zinc: saturated glycine solution (Penta s.r.o., Prague, Czech Republic).Lead: 1500 g of ammonium acetate (CH_3_COONH_4_) (Lach-Ner, s.r.o., Neratovice, Czech Republic) with demineralized water to make up 1000 mL (saturated solution) or 1% HCl.

All pickling baths operated at laboratory temperature and pickling always took 1 min. Thorough rinsing with demineralized water and ethanol, and drying, followed.

The accuracy of the determination of corrosion rate by mass changes depends mainly on the accuracy of weighing. Ultra-microbalances should ideally be able to determine mass to a tenth of a microgram. Since each mass determination is based on four weighings (Equation (2)), and individual errors can add up, the accuracy of the mass-change determination should be expected to be in the order of micrograms. If a coupon has an exposure area of a few square centimeters, then the accuracy of the corrosion rate determination is in milligrams per square meter per year of exposure (mg m^−2^ a^−1^).

#### 2.3.4. Determination of Corrosion Rate by Electrochemical Reduction

Samples that were not subjected to corrosion rate determination by mass loss, and thus were not pickled, were used to determine the electric charge required for the electrochemical reduction of corrosion products. The procedure is well-established for the determination of corrosion products on silver and copper after exposure to indoor atmosphere and is also described in ISO 11844-2. The applicability of the technique has also been tested for lead coupons, since some lead corrosion products are known to be electrochemically reducible [22,32].

The electrolytic cathodic reduction was performed in a 0.1 mol·L^−1^ KCl deaerated solution (Penta s.r.o., Prague, Czech Republic) at a constant current density of 125 μA·cm^−2^ using a Zahner Zennium E electrochemical workstation (ZAHNER-elektrik GmbH & Co. KG, Kronach, Germany). The change of potential of the specimen was monitored during the electrolytic reduction of the corrosion products. The corrosion rate *r_er_* (mg·m^−2^·a^−1^) is given by Faraday’s law, as shown by
(4)$$r_{er}=\frac{i\cdot t_{red}\cdot M}{n\cdot F\cdot t}$$
where *i* is the current density (mA·m^−2^); *t_red_* is the total time for reduction of corrosion products (s); *M* is the relative molecular mass (g·mol^−1^) (107.9 for silver, 63.5 for copper, and 207.2 for lead); *n* is the valance state (1 for silver, 1 or 2 for copper, 2 for lead); and *F* is Faraday’s constant, 96,485 C·mol^−1^.

The accuracy of determining the corrosion rate depends primarily on the accuracy of reading the corrosion product reduction time. The maximum error in the reading is at the level of seconds. An error in determining the reduction time of 10 s means an error in determining the corrosion rate of approximately 10 mg·m^2^·a^−1^.

#### 2.3.5. Color Change

During exposure, the color and lightness changes on the exposed silver, copper, zinc, and lead coupons in the CIE L*a*b* color space (CIELAB) were monitored regularly using a portable Konica Minolta CM-700d spectrophotometer (aperture 3 mm) (Konica Minolta GmbH, Munich, Germany). The color was measured for each metal on 10 coupons; 2 measurements were performed from each side of the coupon in specular component-included and -excluded (SCI and SCE) modes. From the changes in the parameters L*, a*, and b*, which represent color changes in the black/white (i.e., lightness), red/green, and yellow/blue ranges [33], the total color change was calculated according to Equation (5). The results for both modes showed the same trends. For this paper the specular component-included (SCI) mode was chosen, which measures the actual color of the coupon without the influence of the surface condition, i.e., regardless of how smooth or glossy the surface is.
(5)$$\Delta E^{*}=\sqrt[{}]{{\left(\Delta a^{*}\right)}^{2}+{\left(\Delta b^{*}\right)}^{2}+{\left(\Delta L^{*}\right)}^{2}}$$

Here, Δ*a**, Δ*b**, and Δ*L** represent the difference between the average value of the parameter measured on 10 coupons of a given metal before and after exposure.

## 3. Results

### 3.1. Exposure Conditions

Average monthly temperature and relative humidity data, with error bars showing minimum and maximum values, are displayed for sites with complete climatic control (locations 8–10, Figure 2), heated sites with climatic data varying with the seasons (locations 1–4, 6, 7, and 11–13, Figure 2, Figure 3 and Figure 4), and completely uncontrolled sites where the data strongly depended on external conditions (locations 14–16, Figure 5). Location 5, an open bookcase in a monastery library, is specific. The temperature also follows seasonal changes, but the relative humidity remains stable. The average values of temperature and relative humidity are shown in Figure 6. It should be noted that the datasets used to calculate the average values are not equally large because the records are not complete for all sites due to the limited availability of loggers.

Missing values in Table 1 correspond to locations where passive samplers did not fit (archival box, metallic film cassette). Table 1 shows the apparent disagreement between the results of acetic acid concentration measurements by the two techniques. However, it should be considered that the results from the passive samplers are long-term averages calculated from several 30-day readings, whereas the results from the PDTs are essentially instantaneous values, as they were available in 1 to 10 days. Instantaneous values are obviously influenced by ambient conditions and the time of the year. Higher concentrations are recorded during periods when the temperature is elevated at a given location, which increases VOA emissions. When comparing values measured during the period corresponding to the start of coupon exposure, the values from passive samplers exposed for the initial 30 days are closer to the instantaneous values from PDTs (Table 1).

In terms of exposure conditions, relatively high corrosivity can be expected at sites 1, 2, and 4 and probably also at 5 and 12. High concentrations of acetic acid and high relative humidity have been observed at these sites for certain periods of time. Although the acetic acid content at sites 14, 15, and 16 was not high (as the spaces are well ventilated by broken windows), the high average relative humidity, and especially the extreme fluctuations in relative humidity, may also be a reason for the high corrosion aggressiveness. Sites 8, 9, 10, 11, and 13 should be non-aggressive due to the low acetic acid content and the control of relative humidity, which is kept low and stable.

A significant effect of VOA concentration and relative humidity on lead corrosion rate was shown in similar exposure tests in indoor museum environments [34]. Since there is always more than one factor influencing the corrosion rate in real environments, the presence of other atmospheric corrosion stimulators has been checked by means of PDTs. Sulphane and ammonium content always remained below the detection limits of 0.01 and 0.5 ppm, respectively, and formaldehyde was detected in locations 2, 3, 4, 5, 6, and 9 at 0.1 ppm at maximum, i.e., slightly above the detection limit of 0.03 ppm.

### 3.2. Corrosion Rate by Means of Mass Increase of Metal Coupons

In 2020, ISO 11844 was revised. The revision of the first part consisted in introducing lead as a standard metal for testing the corrosion aggressiveness of indoor atmospheres with specific reactivity to volatile organic acids. The revision of the second part corrected an error in the calculation of the mass of the corrosion coupon determined using a reference sample on an ultra-microbalance. The corrected calculation is given in this paper (Equation (2)).

It should be noted that determining the corrosion rate based on mass increase requires high precision and the provision of a high-quality ultra-microbalance in very stable conditions. For example, to determine the IC2 corrosion class for silver, a mass increment of hundredths of a milligram must be reliably detected. In our measurements, weighing at this order of magnitude was already burdened with a large error. When determining the mass of the sample relative to the reference sample according to relation (Equation (2)), the errors of individual weighings add up. Despite the use of an ultra-microbalance, it was not possible to determine the mass changes with the required accuracy. Very often the determination of the corrosion increment resulted in a negative value. To eliminate inaccuracies in weighing with the ultra-microbalance, it helped to reduce the sample size from the original standard recommended size of 10 × 50 mm to 10 × 20 or 10 × 25 mm. Smaller samples are more stable on the ultra-microbalance plate and the reading stabilizes more quickly.

The results of the corrosion rate determination by the mass increase method are shown in Figure 7. In most cases, the highest corrosion rate was detected for lead. The correlation between the increased corrosion rates of the different metals is not apparent at first glance.

### 3.3. Corrosion Rate by Means of Mass Loss of Metal Coupons

The rate of corrosion loss was also evaluated in accordance with the revised ISO 11844 standard. The mass loss of a blank unexposed specimen, which is treated in the same way and whose mass loss is due only to the dissolution of the metal in the pickling bath, is subtracted from the mass change of the exposed coupon after a single pickling cycle (Equation (3)). This is done to eliminate the error caused by including the mass of dissolved metal in the corrosion loss during exposure. Obviously, this procedure can work well if the corrosion products in the pickling bath are removed very quickly, which is the case for less corroded samples. Corrosion products that take the entire pickling period to remove protect the base metal from dissolution in the pickling bath. Subtracting the mass loss of the blank then leads to an underestimation of the corrosion rate. This must be considered when interpreting the results.

From Figure 8, it can be seen that the mass loss method highlights the difference in corrosion rate between lead and other metals, which is expected given the relatively high molar mass of lead compared to the others. While the mass increase is caused by the binding of light elements (mainly oxygen, sulfur, carbon, and hydrogen), the mass loss reflects the loss of atoms with different high molar masses. As in the case of mass increase, sites 5 and 12, where high concentrations of acetic acid were detected, show the highest corrosion rate of lead. Thus, the high sensitivity of lead to volatile organic acids is already evident here. Zinc, which was originally included in ISO 11844 because of its expected sensitivity to VOAs, does not show an increased corrosion rate at these sites. In contrast to the corrosion rate from mass increase, a correlation between corrosion rates of different metals can be observed in many cases.

### 3.4. Corrosion Rate by Means of Electrochemical Reduction of Corrosion Products on Metal Coupons

The electrochemical reduction of corrosion products is an independent method to determine the amount of silver and copper in electrochemically reducible corrosion products. It is based on the polarization of exposed metal samples with a constant cathodic current. The termination of the corrosion product reduction is readily identified by a significant drop in potential. ISO 11844 prescribes a procedure for electrochemical galvanostatic reduction of silver and copper corrosion products. The electrolyte is a solution of KCl at a concentration of 0.1 mol·L^−1^. Coupons of silver, copper, and lead were evaluated by this procedure in this study. It has been shown that even lead exposed to an indoor atmosphere is covered with easily reducible corrosion products, and the passed charge was also used to evaluate the corrosion loss in the case of lead. In converting the passed charge to the amount of corroded lead, it was assumed that lead is oxidized to the oxidation state +II, while silver and copper are oxidized to +I.

The corrosion rates of silver, copper, and lead at each site, calculated by the passed charge during the galvanostatic reduction of the exposed coupons, are shown in Figure 9. It is rather surprising how well the corrosion rates of lead correspond with those determined by mass loss (Figure 8). Not only do the values agree within an order of magnitude, but they also show the same trend when compared between sites. The agreement for silver and copper is not as good, which may be due to the underestimation of the corrosion rate, as described in Section 3.3, or the fact that in some cases two plateaus were observed in the galvanostatic reduction curves, indicating the presence of copper corrosion products in oxidation state +II. In this case, the copper corrosion rates plotted in Figure 9 could be overestimated. It is clear that silver follows the trend of corrosion rates better between sites and is therefore more reliable than copper in this respect.

### 3.5. Color Change

The color change, ∆*E**, was monitored regularly during the exposure, but only the one-year data are presented here. From Figure 10, it is clear that lead showed the greatest overall color change after one-year exposure, with three exceptions, localities 2, 3, and 15. Lead is known for its rapid darkening even under relatively non-aggressive conditions. Similar color changes also occur on copper and silver, which also darken in indoor atmospheres. Zinc shows the least profound color change. Given the typically white or grayish corrosion products of zinc and the silvery color of zinc itself, this is not surprising.

The main cause of the change in lead color is darkening. This is evident in Figure 11, where positive values on the vertical axis represent decreased lightness, i.e., darkening. The drop in lightness corresponds almost entirely to the overall color change ∆*E** for lead. For silver and copper, other color components also contributed to the color change.

## 4. Discussion

### 4.1. Comparison of Techniques for Corrosion Rate Determination

All of the above techniques for determining corrosion rates are quite laborious, time-consuming, and meticulous. In principle, they are supposed to give the same results, although it is known that each is burdened with errors. Figure 12 compares the corrosion rate determined from mass increase by ultra-microbalance with that calculated from the charge required for the electrochemical reduction of corrosion products. Indeed, the former parameter is the mass of elements from the environment bound in corrosion products (mainly oxygen, carbon, and hydrogen, and possibly some sulfur or nitrogen), whereas the latter is the mass of oxidized metal. It is thus clear that datasets for particular metals need to be separated, as the molar masses of Cu, Ag, and Pb are different. Even in particular datasets, some deviations are expected due to differences in corrosion product composition. Still, the fact that there is virtually no agreement in the results, as shown in Figure 12, is surprising. This may be primarily due to the high inaccuracy of the mass increase determination by ultra-microbalance.

Significantly better agreement is observed when comparing the mass loss rate determined by ultra-microbalance with that converted from the charge required for the electrochemical reduction of corrosion products, if we disregard the results for copper (Figure 13). In contrast to silver, with copper it is less predictable what corrosion product is present on the surface after exposure. It is also possible that the corrosion rate determined by the mass loss method underestimates the relatively high loss of the blank during pickling. The values for lead are in reasonably good agreement. For silver, the values for mass loss determined by the ultra-microbalance are optically higher in logarithmic coordinates, but since these are mostly in the high tens versus the first hundred, the difference is not significant.

### 4.2. Comparison of Corrosion Rate with Color Changes

The formation of corrosion products during exposure to atmospheric conditions naturally leads to a darkening of the corrosion coupons or a color change. The degree of such changes should be proportional to the thickness of the corrosion layer. The results of the phase analysis by XRD showed that after one year of exposure, the corrosion products present in any sample could not be reliably identified. The corrosion products were studied by means of XRD, but only in the case of lead. Oxide-, carbonate-, acetate-, and formate-based corrosion products were identified, but always inconclusively. A more extensive analysis is planned for a series of samples that are still exposed and whose evaluation is planned after 3 years of exposure. We anticipate using XRD, FTIR, and XPS. The authors plan to publish the results of the 3-year exposure in a future paper.

The change in appearance can be monitored continuously and non-destructively on the corrosion coupons using the CIELab system. The change in overall color of the silver, copper, and lead coupons after 1 year of exposure was compared to the corrosion rate determined from the mass loss in Figure 14. While the differences in color change between coupons are within units, the corrosion rates vary by orders of magnitude. Therefore, corrosion rates are shown on a logarithmic scale, and any linear regression curves naturally do not have the shape of a straight line in logarithmic coordinates. For copper, although the sample with the lowest corrosion rate changed its color the least during exposure and the sample with the highest corrosion rate changed the most, the scattering of other values is too large to consider copper color change as a useful tool for assessing corrosion rates. In the case of silver and lead, a nearly linear trend can be observed. The regression curve includes all experimental points in the case of silver; in the case of lead, four points lying completely outside the trend were ignored. These points correspond to locations 13 (∆*E** = 20.9), 12 (22.9), 11 (18.9), and 4 (14.5). There is no explanation for the out-of-trend point for location 13. This is a location with generally normal conditions in terms of temperature, relative humidity, and acetic acid concentration. Sites 4, 11, and 12 are characterized by high lead corrosion rates, as identified by mass loss. Crystals of white corrosion products, presumably carbonate- or acetate-based, were observed on lead samples at these locations after exposure. Since lightness is the main parameter affecting the overall color change of lead, it is obvious that white corrosion products can distort the information provided by the color change. A comparison of the color change of lead after 1 and 3 months of exposure suggests that it is more appropriate to use color change values after 3 months to assess the aggressiveness of the indoor atmosphere toward lead. The annual exposure breaks the linear dependence of the color change on the corrosion rate. Thus, based on the 3-month results for lead and the annual results for silver, it can be argued that quantification of color change can serve as a simple tool for assessing the corrosiveness of an indoor atmosphere.

### 4.3. Effect of Exposure Conditions on the Corrosion Rate of Metal Coupons

The first environmental parameter that should be assessed in relation to lead corrosion is the content of volatile organic acids in the atmosphere. To assess the effect of VOA concentration on the corrosion rate of lead, acetic acid concentration values determined by PDTs at the beginning of the corrosion coupon exposure were considered. Against these data, corrosion rates determined using mass loss after one year of exposure were plotted. These were taken as the most relevant according to the above comparisons.

The detected concentrations of acetic acid are not expected to affect the corrosion rate of silver and copper (Figure 15). The highest corrosion rates of lead were observed at sites with high acetic acid concentrations. However, two different trends can be distinguished, one in which the corrosion rate does not increase much with increasing acetic acid concentration (Pb1 in Figure 15), and one in which the corrosion rate is directly proportional to the acetic acid content with a directive value close to 1 (Pb2 in Figure 15). No trend is observed at acetic acid concentrations below 1000 µg·m^−3^, which is the majority of sites. The slopes of both of these dependencies are several-fold lower than those observed in the study of Ryhl-Svendsen [34] when converted to the same exposure time and expressed in ppb of acetic acid. This can be explained by the shorter exposure time of 3 months in Ryhl-Svendsen’s study. The corrosion rate usually decreases with time, and here the corrosion rates after 1 year and 3 months of exposure were compared. Furthermore, sites with acetic acid content higher than 200 ppb (530 µg·m^−3^) were not included in the Ryhl-Svendsen study. As seen in Figure 15, the corrosion rate dependency could be steeper at lower acetic acid contents. However, our data do not allow us to apply linear regression in this region.

Clearly, the acetic acid concentration in air is not the only factor that determines air corrosivity toward lead. With extreme acetic acid contents, the corrosion rate is obviously high despite the fact that the exposure temperature is stable and the relative humidity is low (location 12), and even that environment should be rated as extremely aggressive. Furthermore, in other locations where elevated corrosion rates above 3000 mg·m^−2^·a^−1^ were observed (locations 4 and 11), the climatic conditions were stable and moderate in terms of relative humidity. However, the higher overall corrosion rate may have been caused by the unstable and high temperature during the summer months (above 25 °C, temporarily above 30 °C), which probably facilitated acetic acid release and led to elevated concentrations in these cryptoclimates. Similar temperature increases occurred at other locations (1, 2, 3, and 5), but these locations were characterized by a higher air exchange and apparently did not experience such significant increases in VOA concentration. The resulting average corrosion rate then did not exceed 2500 mg·m^−2^·a^−1^. The group of sites with slightly elevated concentrations of acetic acid, on the order of hundreds of micrograms per cubic meter, and corrosion rates below 2500 mg·m^−2^·a^−1^, includes, in addition to location 3, locations 6 and 7. Despite the presence of VOAs, the corrosion rate of lead remained at a relatively low level.

The last characteristic group consists of locations 8–10 and 13–16. These locations are characterized by acetic acid contents on the order of tens of micrograms per cubic meter and lead corrosion rates below 2500 mg·m^−2^·a^−1^. Locations 8, 9, 10, and 13 are archives and depositories with effective air filtration or very good temperature and relative humidity control. In the case of location 10, the low concentration of VOAs is due to material with an alkaline reserve (modern archive box). Locations 14, 15, and 16 are well ventilated, as they are churches in poor condition with broken windows. The low lead corrosion rates at these sites demonstrate that without the presence of approximately 200–400 µg·m^−3^ of acetic acid, an increased corrosion rate is improbable even if there were strong variations in temperature and relative humidity. On the other hand, high corrosion rates can be achieved even at relatively low instantaneous VOA concentrations and low relative humidity when a temporary increase in ambient temperature results in the release of higher amounts of VOAs. It is thus important to prevent temperature fluctuations, especially in unventilated spaces (cryptoclimates) for the protection of lead collection objects.

### 4.4. Corrosivity Classes

The corrosivity classification of indoor atmospheres according to ISO 11844 is widely accepted (Table 2). Table 3 shows the corrosivity categories of the exposure locations for silver based on mass loss and electrochemical reduction. This classification is based on the corrosion rate limits set by ISO 11844-1 [29]. In terms of corrosivity, all selected locations have very low indoor corrosivity (IC1), except locations 14 and 15. From the mass loss data, these locations have low indoor corrosivity (IC2), but it should be noted that the corrosion rate is very close to the limit between IC1 and IC2. Electrochemical reduction ranked them as IC1 sites.

In contrast, according to the corrosion rate of lead, all locations fall into corrosivity category IC5, or the corrosion rate is even higher than the upper limit of this category. This indicates that the limits for the corrosivity categories are not realistically set in the current version of ISO 11844-1. Based on the results of one-year exposure under realistic conditions, new limits for the corrosivity classes were proposed (Table 2), and the locations were subsequently classified into these classes (Table 3). These suggest that the very low indoor corrosivity class (IC1) for lead corresponds to the environment inside an archive box with an alkaline reserve, with effective air filtration and controlled temperature and relative humidity in the archive, or a museum depository with similarly controlled conditions. The only environment in the IC5 category (very high corrosivity) is the interior of the metallic case, which houses acetic acid-releasing cellulose acetate film. The corrosion rate boundary between IC2 and IC3 was chosen to separate sites with sources of VOAs and controlled air quality (locations 6, 8–10) and sites with good air exchange (15, 16) from sites with sources of VOAs without air quality control (1–3, 5, 7). IC4 was reserved for sites in the latter group where elevated corrosion rates were found (locations 4, 11).

## 5. Conclusions

The procedure for determining the weight of corrosion coupons according to ISO 11844 is difficult to follow. Although an error in the weight calculation was corrected in the new version of the standard issued in 2020 at the suggestion of the author’s team, the method for determining the corrosion rate from mass increase is difficult to apply and is burdened with a large experimental error. Even the corrosion rate from mass loss can be subject to a large error if the procedure set out in ISO 11844 is applied to relatively heavily corroded samples. Electrochemical galvanostatic reduction has shown good agreement with the mass loss method for silver as well as lead. Lead, which is newly introduced to ISO 11844 as a standard metal for determining the corrosivity of indoor atmospheres, can also be evaluated by electrochemical reduction, provided that the exposure does not produce irreducible corrosion products.

The CIELab color-change measurement has proven to be a simple and easy-to-apply method for monitoring the corrosivity of indoor atmospheres. The rate of change in lead color or lightness corresponds well with the corrosion rate determined by mass loss. A more reliable response is provided by the determination of color change after 3 months of exposure than after 1 year.

At the selected sites, lead always showed significantly higher corrosion rates than silver, copper, and zinc, and they usually corresponded to the expected corrosivity considering the presence of volatile organic acid sources and acetic acid content. This demonstrates the higher sensitivity of lead to the presence of acetic acid in the air and justifies its inclusion in the ISO 11844 standard as a metal for monitoring the corrosivity of indoor atmospheres. However, the relationship between the lead corrosion rate and acetic acid content is not as straightforward as previously published [34]. It is probably affected by relative humidity and temperature fluctuations, which, especially in cryptoclimates, could temporarily increase the content of VOAs, which might not be revealed by the occasional monitoring of their concentrations. Without the presence of approximately 200–400 µg·m^−3^ of acetic acid, an increased lead corrosion rate is improbable even if there were strong variations in temperature and relative humidity. Nevertheless, from the preventive conservation point of view, it is advisable to prevent temperature fluctuations in order to protect lead collection objects, especially in unventilated spaces (cryptoclimates), where the acetic acid content may exceed the range mentioned above.

According to the currently valid lead corrosivity categories, corrosion rates in most localities in this study were above the highest category, IC5. Thus, the current classification was found to be unrealistic. New intervals have been proposed and will be considered for the next revision of ISO 11844-1.

## Figures and Tables

**Figure 1 materials-15-00639-f001:**
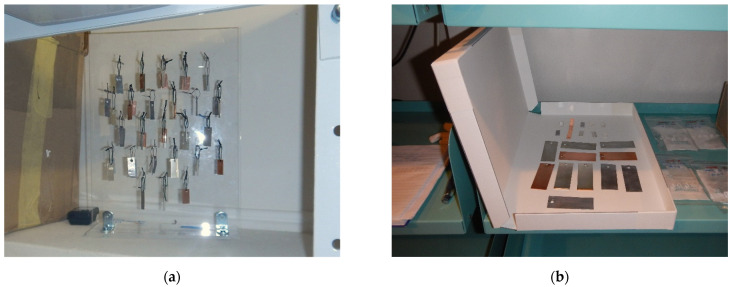
Corrosion coupons: (**a**) fixed to stand at location 13; (**b**) placed in modern archive box at location 10.

**Figure 2 materials-15-00639-f002:**
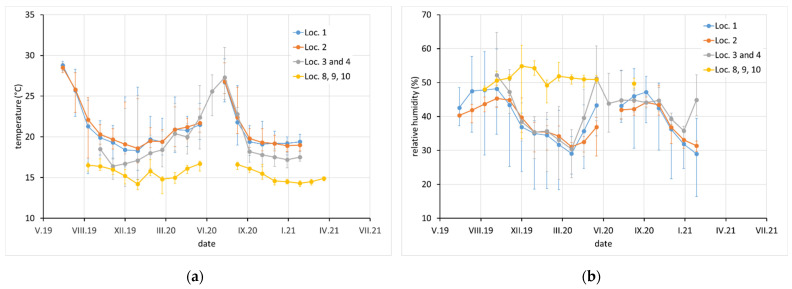
(**a**) Temperature and (**b**) relative humidity record at locations 1, 2, 3, 4, 8, 9, and 10. Points indicate average monthly values and error bars indicate monthly minimum and maximum.

**Figure 3 materials-15-00639-f003:**
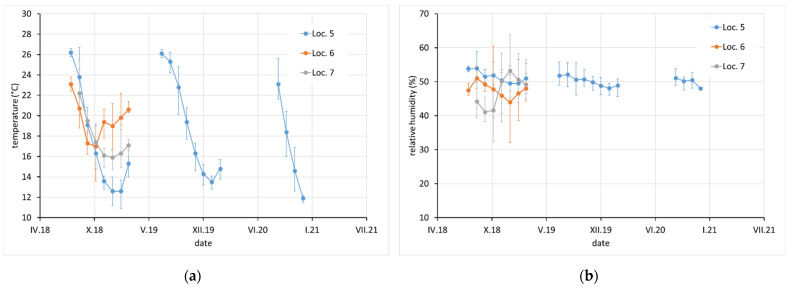
(**a**) Temperature and (**b**) relative humidity record at locations 5, 6, and 7. Points indicate average monthly values and error bars indicate monthly minimum and maximum.

**Figure 4 materials-15-00639-f004:**
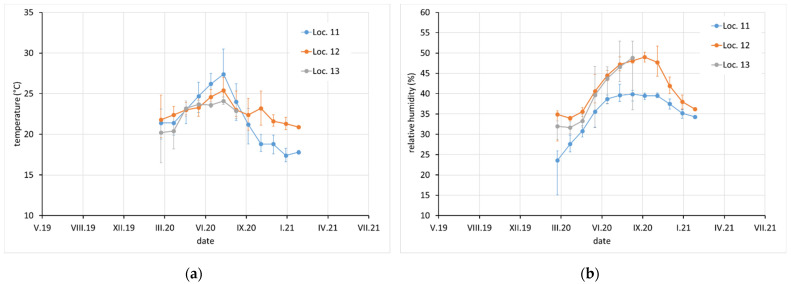
(**a**) Temperature and (**b**) relative humidity record at locations 11, 12, and 13. Points indicate average monthly values and error bars indicate monthly minimum and maximum.

**Figure 5 materials-15-00639-f005:**
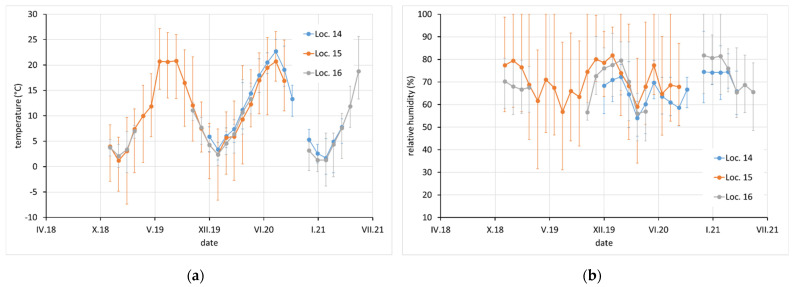
(**a**) Temperature and (**b**) relative humidity record at locations 14, 15, and 16. Points indicate average monthly values and error bars indicate monthly minimum and maximum.

**Figure 6 materials-15-00639-f006:**
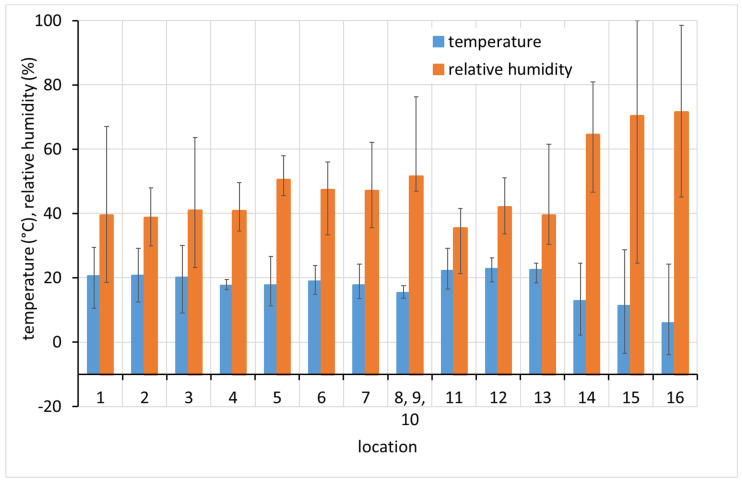
Overall average temperature and relative humidity values. Columns indicate average values and error bars indicate minimum and maximum values from complete dataset.

**Figure 7 materials-15-00639-f007:**
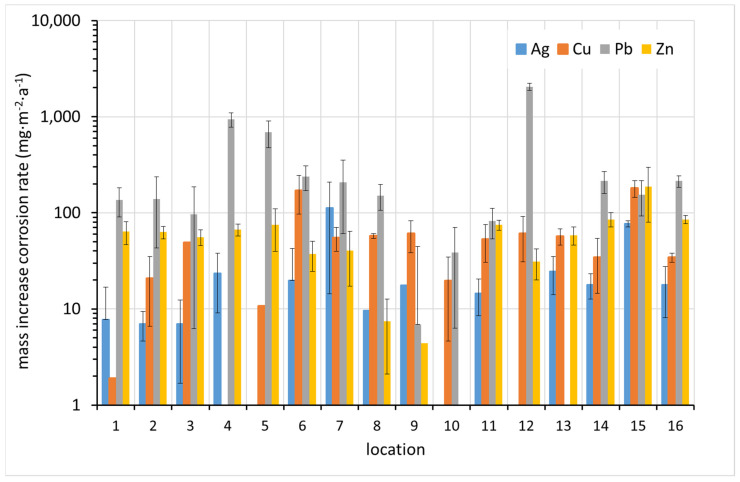
Corrosion rate of coupons based on mass increase (average values from up to six replicates supplemented with standard deviation error bars where applicable).

**Figure 8 materials-15-00639-f008:**
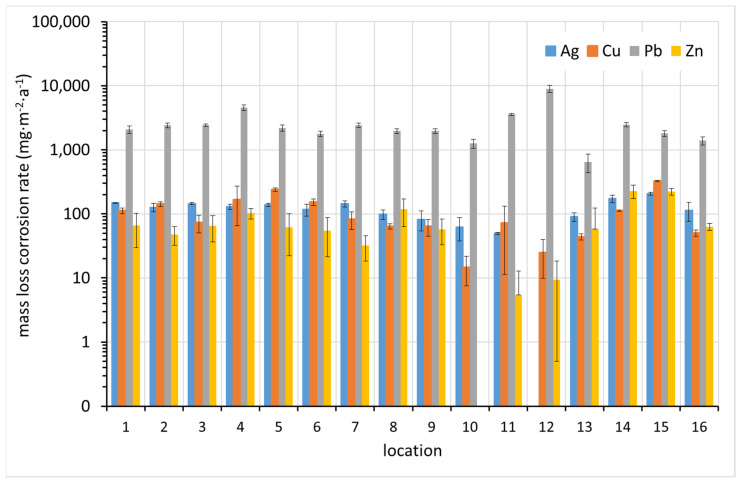
Corrosion rate of coupons based on mass loss (average values from up to three replicates supplemented with standard deviation error bars where applicable).

**Figure 9 materials-15-00639-f009:**
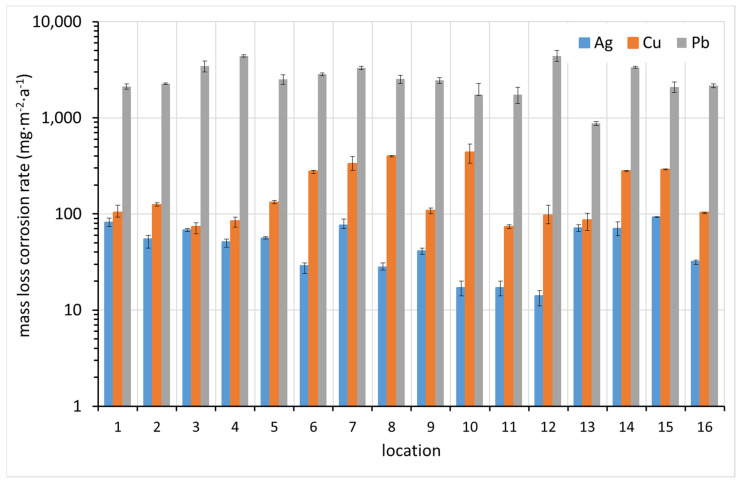
Corrosion rate from mass loss of corrosion coupons calculated from galvanostatic reduction charge (average values from up to three silver and copper replicates and two lead replicates; error bars represent minimum and maximum values of results).

**Figure 10 materials-15-00639-f010:**
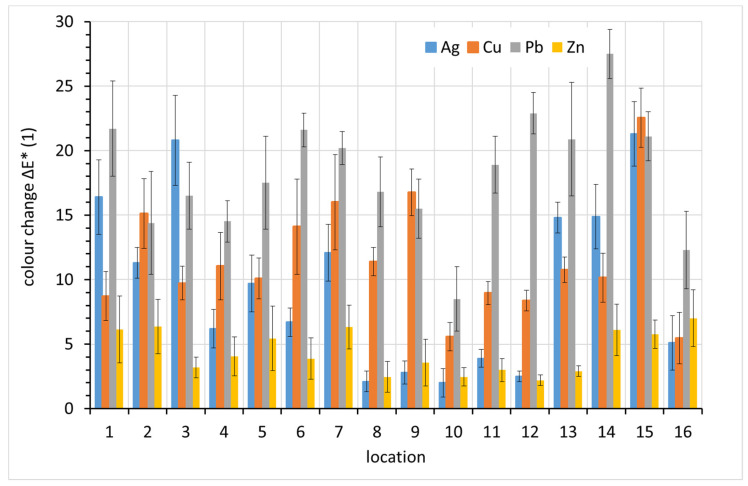
Color change of coupons after a year of exposure (average values from 10 recordings supplemented with standard deviation error bars).

**Figure 11 materials-15-00639-f011:**
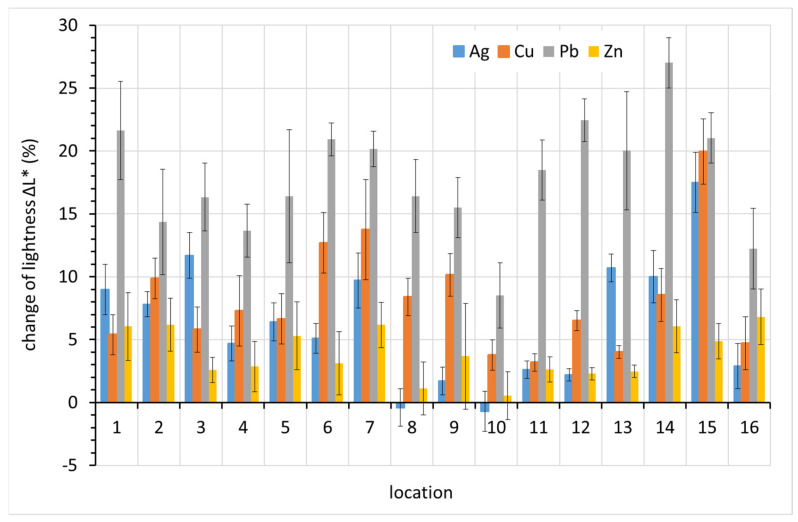
Change of lightness Δ*L**, one of three color change components after a year of exposure of metal coupons (average values obtained from 10 recordings on different coupons before and after exposure and supplemented with standard deviation error bars). Higher value of lightness change means darker appearance.

**Figure 12 materials-15-00639-f012:**
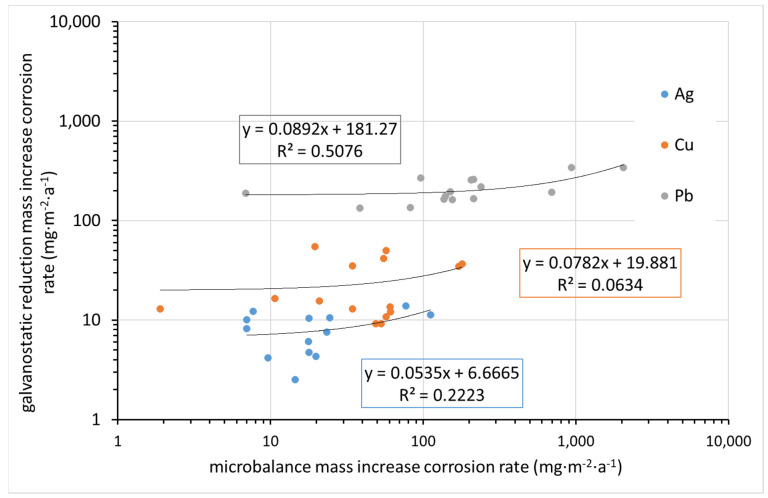
Comparison of corrosion rate after one year of exposure as obtained from microbalance mass increase and electrochemical galvanostatic reduction (in regression equations, y indicates galvanostatic reduction mass loss corrosion rate and x indicates microbalance mass increase corrosion rate).

**Figure 13 materials-15-00639-f013:**
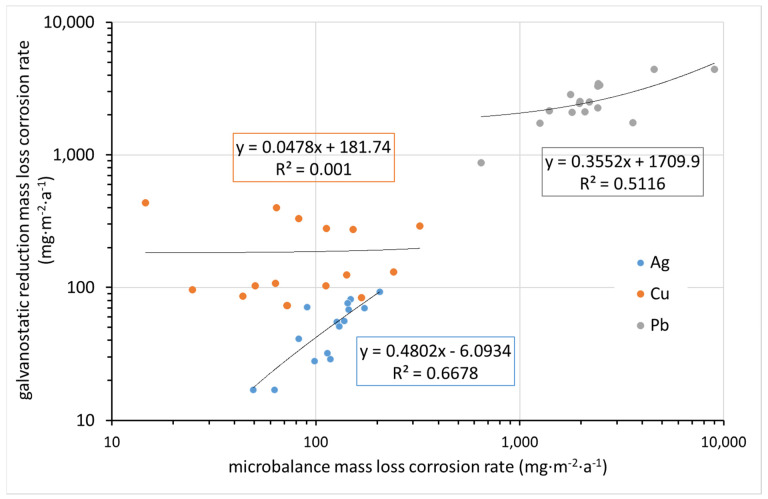
Comparison of corrosion rate after one year of exposure as obtained from microbalance mass loss and from electrochemical galvanostatic reduction (in regression equations, y indicates the galvanostatic reduction mass loss corrosion rate and x indicates the microbalance mass loss corrosion rate).

**Figure 14 materials-15-00639-f014:**
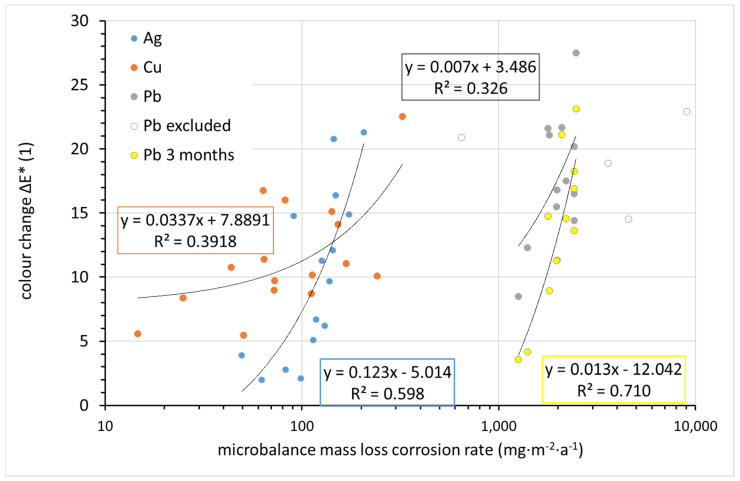
Comparison of corrosion rate as obtained from microbalance mass loss with color change (in regression equations, y indicates color change and x indicates the microbalance mass loss corrosion rate).

**Figure 15 materials-15-00639-f015:**
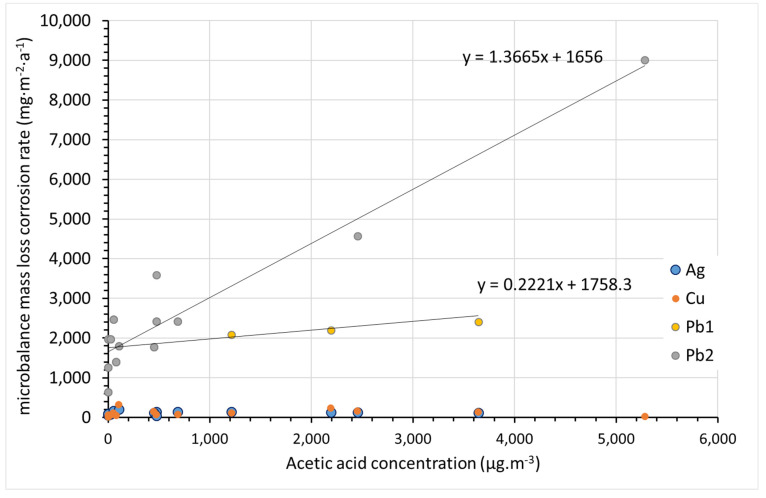
Influence of atmospheric acetic acid concentration on corrosion rate of metals. Acetic acid concentration was quantified by PDTs at start of coupon exposure. (Note: Lead samples from Pb1 and Pb2 datasets do not differ in principle, but follow a different trend of corrosion rate vs. VOA content.).

**Table 1 materials-15-00639-t001:** Acetic acid concentrations.

Location	Start of Corrosion Coupon Exposure	Overall Exposure Period of Passive Samplers	Acetic Acid Concentration (μg·m^−3^)		
			Passive Samplers, Overall Exposure Average	PDT, Exposure Start	Passive Samplers, Exposure Start
1	26 July 2019	February 2020–March 2021	367 ± 37	1214	965 ± 38
2	26 July 2019	February 2020–March 2021	665 ± 35	3643	2324 ± 128
3	7 October 2019	October 2020–October 2021	468 ± 37	475	268 ± 12
4	7 October 2019	October 2020–October 2021	1778 ± 81	2455	1206 ± 107
5	26 July 2019	December 2019–December 2020	291 ± 14	2191	570 ± 18
6	30 August 2019	December 2019–October 2020	193 ± 16	449	309 ± 26
7	30 August 2019	February 2020–October 2020	240 ± 18	686	279 ± 3
8	11 September 2019	December 2019–November 2021	313 ± 19	<DL	62 ± 5
9	11 September 2019	-	-	26	-
10	11 September 2019	-	-	<DL	-
11	5 March 2020	June 2021–October 2021	2237 ± 67	475	NA
12	5 March 2020	-	-	5280	NA
13	5 March 2020	June 2021–October 2021	445 ± 37	<DL	NA
14	29 November 2019	May 2021–November 2021	403 ± 25	53	NA
15	1 November 2019	November 2019–May 2020	102 ± 4	106	52 ± 5
16	29 October 2019	June 2021–November 2021	250 ± 26	79	282 ± 6

DL, detection limit.

**Table 2 materials-15-00639-t002:** Corrosivity categories of indoor atmosphere based on corrosion rate measurements by mass loss determination of standard specimens according to ISO 11844 and new proposal based on results of this study.

Corrosivity Category	Corrosion Rate r_corr_ (mg·m^−2^·a^−1^)		
	Ag (ISO 11844)	Pb (ISO 11844)	Pb (New Proposal)
IC1	*r_corr_* ≤ 170	*r_corr_* ≤ 40	*r_corr_* ≤ 1300
IC2	170 < *r_corr_* ≤ 670	40 < *r_corr_* ≤ 150	1300 < *r_corr_* ≤ 2000
IC3	670 < *r_corr_* ≤ 3000	150 < *r_corr_* ≤ 400	2000 < *r_corr_* ≤ 3000
IC4	3000 < *r_corr_* ≤ 6700	400 < *r_corr_* ≤ 700	3000 < *r_corr_* ≤ 6000
IC5	6700 < *r_corr_* ≤ 16,700	700 < *r_corr_* ≤ 1600	6000 < *r_corr_* ≤ 20,000

**Table 3 materials-15-00639-t003:** Corrosivity categories of exposure locations.

Location	Ag (Mass Loss; ISO 11844)	Ag (Electrochemical Reduction; ISO 11844)	Pb (Mass Loss; ISO 11844)	Pb (Mass Loss; New Proposal)
1	IC1	IC1	>IC5	IC3
2	IC1	IC1	>IC5	IC3
3	IC1	IC1	>IC5	IC3
4	IC1	IC1	>IC5	IC4
5	IC1	IC1	>IC5	IC3
6	IC1	IC1	>IC5	IC2
7	IC1	IC1	>IC5	IC3
8	IC1	IC1	>IC5	IC2
9	IC1	IC1	>IC5	IC2
10	IC1	IC1	>IC5	IC1
11	IC1	IC1	>IC5	IC4
12	-	IC1	>IC5	IC5
13	IC1	IC1	IC5	IC1
14	IC2	IC1	>IC5	IC3
15	IC2	IC1	>IC5	IC2
16	IC1	IC1	>IC5	IC2

## Data Availability

Not applicable.

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
