# Peer review of "Lead Corrosion and Corrosivity Classification in Archives, Museums, and Churches"

_materials, 2022, doi:10.3390/ma15020639_

Round 1
Reviewer 1 Report
Have been done any characterization of the corrosion products? There is any relationship between corrosion product composition and colour changes?
Line 603 cross reference is missing
Author Response
Point 1: Have been done any characterization of the corrosion products? There is any relationship between corrosion product composition and colour changes?
Response 1: Dear reviewer, thanks a lot for your comments. The analysis of corrosion products was not given much attention at this stage of the work. The corrosion products have been studied by means of XRD but only in the case of lead. As the layer of corrosion products is very thin after one year of exposure, they are difficult to identify by this technique. Certainly, their quantification is unreliable. A more extensive analysis is planned for a series of samples that are still exposed and whose evaluation is planned only after three years of exposure. We anticipate the use of XRD, FTIR and XPS. The authors plan to publish the results of the three-year exposure in a future paper. On page 16, this explanation has been added and preliminary XRD results for lead have been added.
Point 2: Line 603 cross reference is missing.
Response 2: Thank you for the notice. All cross references have been fixed. The cross reference no. 29 has been added.
Reviewer 2 Report
Notes to the Authors:
1. The method of determining the accuracy of the determination of the corrosion rate should be provided
2. On Figs 7 ÷ 11, the error bars in determining the corrosion rate should be marked
3. The unit in Figs 12, 13, 14, 15 should be corrected in mg.m-2.a-1 per mg·m-2·a-1
4. In Fig. 14 the regression curve should be presented for the Cu
5. In Fig 14 the regression curve is parabolic and the regression curve is linear - these relationships should be explained; you also need to explain what is y and what is x in regression formulas
6. Line 603 removing the text Error! Reference source not found ..
7. In Figures 12 and 13, add a regression curve
8. Please explain how the Corrosion rate rcorr ranges for Pb were determined (new proposal)
9. Corrosivity categories of exposure locations differ significantly for the different metals tested; is there a method to determine the resultant corrosivity category?
10. The authors should present sample microscopic images of the surface and profilograms for the tested samples
Author Response
Point 1: Dear review, thank you for all the comments. The method of determining the accuracy of the determination of the corrosion rate should be provided
Response 1: The accuracy of corrosion rate determination is undoubtedly a key issue. The expected level of accuracy is newly described in Chapters 2.3.3 and 2.3.4.
Point 2: On Figs 7 ÷ 11, the error bars in determining the corrosion rate should be marked
Response 2: Thanks a lot again for the comment. The absence of error bars is, of course, a big mistake.
Error bars representing standard deviation were added to Figure 7, 8, 10 and 11 except in cases where less than three meaningful values were available Figure 7 and 8. Exceptionally, the lower error bar is also omitted, in cases where it interferes with negative values and thus cannot be displayed in logarithmic scale.
The error bars in Figure 9 do not represent standard deviations, there are few values available for that, but they do show the minimum and maximum values. This approach also allows to assess the reliability of the data. The text is supplemented by the relevant commentary marked in yellow.
Point 3: The unit in Figs 12, 13, 14, 15 should be corrected in mg.m-2.a-1 per mg·m-2·a-1
Response 3: The unit has been corrected in all the figures.
Point 4: In Fig. 14 the regression curve should be presented for the Cu
Response 4: The regression curve for copper has been presented, as well as the reliability values R.
Point 5: In Fig 14 the regression curve is parabolic and the regression curve is linear - these relationships should be explained; you also need to explain what is y and what is x in regression formulas
Response 5: The regression curve is not straight but it is because of logarithmic scale of the corrosion rate coordinate. The regression curve for copper has been presented, as well as the reliability values R. An explanation of x and y meanings has been provided in Fig 14.
Point 6: Line 603 removing the text Error! Reference source not found ..
Response 6: All references have been fixed.
Point 7: In Figures 12 and 13, add a regression curve
Response 7: Regression curves have been added.
Point 8: Please explain how the Corrosion rate rcorr ranges for Pb were determined (new proposal)
Response 8: All of the lead corrosion rates determined in this study would be in the highest corrosion rate class or even above the highest corrosion rate limit under the current standard. Yet all exposures occurred in normal environments that are characteristic of the conservation area. This clearly indicated that the corrosion rate limits needed to be modified. The design of new corrosion rate limits for each class is based on a combination of observed corrosion rates and knowledge of local conditions. Class IC1 one includes a location with very carefully controlled air quality (location 13), where no corrosion problems have been observed on stored lead objects for a long time, and a cryptoclimate of a modern archive box made of non-aggressive materials (location 10) and stored in conditions with very carefully controlled air quality. At the opposite extreme is cryptoclimate with acetate film and clearly the highest corrosion rate of lead (location 12). The corrosion rate boundary between class IC2 and IC3 was chosen to separate sites with sources of VOAs but with controlled air quality (locations 6, 8-10) or sites with good air exchange (locations 15 and 16) from sites with sources of VOAs without air quality control (location 1-3, 5, 7). The IC4 category was reserved for sites in this group where elevated corrosion rates were found (locations 4 and 11).
Chapter 4.4 has been added accordingly.
Point 9: Corrosivity categories of exposure locations differ significantly for the different metals tested; is there a method to determine the resultant corrosivity category?
Response 9: The corrosivity of the atmosphere can be determined by different factors for each metal. Corrosion stimulators can have very different effects on different metals. Therefore, the standards for assessing the corrosivity of the atmosphere also include several standard metals by which corrosivity classes are determined. An atmosphere that is aggressive to silver may not be aggressive to lead. Lead was recently included in ISO 11844 because the existing standard metals did not adequately cover the aggressiveness induced by VOAs. We believe that specifying a single category of corrosion aggressiveness is not desirable. On the contrary, the aggressiveness should be evaluated individually for all standard metals and from this it should then be deduced what could possibly account for the increased corrosion aggressiveness.
Point 10: The authors should present sample microscopic images of the surface and profilograms for the tested samples
Response 10: Thank you for the suggestion. Actually, the microscopic images of the surface and profilograms have not been taken after this one-year exposure. A more extensive analysis is planned for a series of samples that are still exposed and whose evaluation is planned after three years of exposure. Detailed corrosion product analysis and microscopic imaging will be done and published afterwards.
Reviewer 3 Report
The authors verified the updates of the standard ISO 11844 in terms of the corrosivity of low-corrosive atmospheres, such as indoor atmospheres of museums, libraries, churches, etc. especially in terms of the introduced lead as a corrosivity indicator, sensitive to contamination with volatile organic acids (mainly acetic acid content). On the basis of the conducted analyzes, they propose a classification correction based on the assessment of lead corrosion under these conditions.
My comments are as follows
- The following statement does not seem to be true to me (line 83-86) ‘However, the objective was not to verify the dependence of lead corrosion rates on climatic conditions and corrosion stimulant content as this is already well documented in other exposure surveys or under laboratory’, since this is the subject of the manuscript.
- The statement in the expression (line 86) ‘to verify the available exposure methods for determining lead corrosion’ is also unclear to me. I think after removing the word 'exposure' the sentence is understandable.
- In general, it seems to me that this paragraph (l. 79-88) needs to be rewrited to be clearer.
- l. 283, it should be ‘mg .m-2 a-1 ‘
- In general, the figures should be more legible, mainly the points on the graphs, their colors should be more definitely different. In Fig. 15, the points for Ag and Cu are difficult to see and distinguish.
- Fig. 14 shows the linear equations which, I guess, describe the curves in the figure, but these curves are not straight lines !!! Please correct this.
- How is Pb1 different from Pb2 in Figure 15? This information should be in the picture.
- l. 603, What it means: Error! Reference source not found.. Please delete it.
- l. 604-605, words: Very and Low should start with a lowercase letter.
These points should be clarified and corrected before further evaluation.
Author Response
Point 1: The following statement does not seem to be true to me (line 83-86) ‘However, the objective was not to verify the dependence of lead corrosion rates on climatic conditions and corrosion stimulant content as this is already well documented in other exposure surveys or under laboratory’, since this is the subject of the manuscript.
Response 1: Dear reviewer, thank you for the comment. We see now that the sentence is confusing. It has been removed.
Point 2: The statement in the expression (line 86) ‘to verify the available exposure methods for determining lead corrosion’ is also unclear to me. I think after removing the word 'exposure' the sentence is understandable.
Response 2: Thanks again. The word “exposure” really does not bring anything useful to the sentence. It has been removed.
Point 3: In general, it seems to me that this paragraph (l. 79-88) needs to be rewrited to be clearer.
Response 3: Hopefully, after the two corrections above, the paragraph has been clarified.
Point 4: l. 283, it should be ‘mg .m-2 a-1 ‘
Response 4: Thanks a lot for the comment. The minus sign has been added.
Point 5: In general, the figures should be more legible, mainly the points on the graphs, their colors should be more definitely different. In Fig. 15, the points for Ag and Cu are difficult to see and distinguish.
Response 5: All the figure (12-15) have been enlarged. In Fig. 15, the points for Ag have been magnified. Hopefully, the hidden points can be seen now more clearly.
Point 6: Fig. 14 shows the linear equations which, I guess, describe the curves in the figure, but these curves are not straight lines !!! Please correct this.
Response 6: The regression curves are not straight but it is because of logarithmic scale of the corrosion rate coordinate.To be more clear to readers, a new sentence has been added to the second paragraph of the chapter 4.2: Then, any linear-regression curves naturally do not have the shape of a straight line in logarithmic coordinates.
Point 7: How is Pb1 different from Pb2 in Figure 15? This information should be in the picture.
Response 7: Lead samples from the Pb1 and Pb2 datasets do not differ in principle, but follow a different trend of corrosion rate vs. VOAs content. This sentence has been added to the title of Figure 15 as a note.
Point 8: l. 603, What it means: Error! Reference source not found.. Please delete it.
Response 8: All references have been fixed.
Point 9: l. 604-605, words: Very and Low should start with a lowercase letter.
Response 9: The capital letters have be chosen because the expressions “Very low indoor corrosivity” and „Low indoor corrosivity” are defined by the ISO standard for the particular corrosivity category. But it is not necessary to use the capital letters in the text. The capital letters have been removed.
Reviewer 4 Report
This work study Lead corrosion and corrosivity classification in archives, museums and churches. It is interest to the researchers in the related area. In general, the paper is innovative. This is a well-written paper that is suitable for publication after the following revisions:
- The English in this paper needs to be greatly improved.
- Abstract is not attractive and should be improved, and please reformulate the abstract in order to clearly show the strengths of this work.
- The introduction is very weak and must be improved. The authors should explain the importance of this work in detail in order to attract the readership of the Journal. It is recommended to cite references Journal of Colloid and Interface Science 585 (2021) 287–301, and Journal of Colloid and Interface Science 582 (2021) 918–931.
4.The resolution of the pictures in the manuscript is too low
- Main findings should also be provided in conclusions.
6.The reference list should be arranged according to the journal requirements.
Based on these, I advise the authors to rectify the above mentioned errors and we hope to re-evaluate the revised manuscript.
Author Response
This work study Lead corrosion and corrosivity classification in archives, museums and churches. It is interest to the researchers in the related area. In general, the paper is innovative. This is a well-written paper that is suitable for publication after the following revisions:
Dear reviewer, thank you for all the comments. All address very important issues. We have done our best to improve the paper accordingly.
Point 1: The English in this paper needs to be greatly improved.
Response 1: The paper has been checked again and some corrections have been introduced
Point 2: Abstract is not attractive and should be improved, and please reformulate the abstract in order to clearly show the strengths of this work.
Response 2: The abstract has been improved. The changes are masked in yellow in the resubmitted document.
Point 3: The introduction is very weak and must be improved. The authors should explain the importance of this work in detail in order to attract the readership of the Journal. It is recommended to cite references Journal of Colloid and Interface Science 585 (2021) 287–301, and Journal of Colloid and Interface Science 582 (2021) 918–931.
Response 3: The introduction has been supplemented with a new text marked in green. We believe that it brings some improvement to the introduction. Thank you for recommending the papers to be cited. Their scope however is quite far from the scope of the submitted paper. We decided not to include the papers “Insight into anti-corrosion nature of Betel leaves water extracts as the novel and eco-friendly inhibitors” and “Papaya leaves extract as a novel eco-friendly corrosion inhibitor for Cu in H2SO4 medium” into to the introduction.
Point 4: The resolution of the pictures in the manuscript is too low
Response 4: The pictures have been inserted again with higher resolution.
Point 5: Main findings should also be provided in conclusions.
Response 5: The conclusions have been expanded a bit with other important findings.
Point 6: The reference list should be arranged according to the journal requirements.
Response 6: All the references have been checked and modified according to the journal requirements.
Round 2
Reviewer 2 Report
Thank you for heeding my suggestions and further explanations. I will recommend publishing the article.
Author Response
Dear reviewer, thank you for helping us to improve the paper. Extensive editing of English has been done by MDPI English Editing in the final version.
Reviewer 4 Report
Accept
Author Response

(The authors gave the same response as above.)
